# Polysaccharides of *Salsola passerina*: Extraction, Structural Characterization and Antioxidant Activity

**DOI:** 10.3390/ijms232113175

**Published:** 2022-10-29

**Authors:** Victoria Golovchenko, Sergey Popov, Vasily Smirnov, Victor Khlopin, Fedor Vityazev, Shinen Naranmandakh, Andrey S. Dmitrenok, Alexander S. Shashkov

**Affiliations:** 1Institute of Physiology of Federal Research Centre “Komi Science Centre of the Urals Branch of the Russian Academy of Sciences”, 167982 Syktyvkar, Russia; 2School of Arts and Sciences, National University of Mongolia, Ulaanbaatar 14201, Mongolia; 3N.D. Zelinsky Institute of Organic Chemistry, Russian Academy of Sciences, 119991 Moscow, Russia

**Keywords:** pectin, polysaccharides, NMR spectroscopy, arabinan, galacturonan (HG), rhamnogalacturonan-I (RG-I), DPPH radical scavenging, phenolic compounds (PCs)

## Abstract

The above-ground part of the *Salsola passerine* was found to contain ~13% (*w*/*w*) of polysaccharides extractable with water and aqueous solutions of ammonium oxalate and sodium carbonate. The fractions extracted with aqueous sodium carbonate solutions had the highest yield. The polysaccharides of majority fractions are characterized by similar monosaccharide composition; namely, galacturonic acid and arabinose residues are the principal components of their carbohydrate chains. The present study focused on the determination of antioxidant activity of the extracted polysaccharide fractions and elucidation of the structure of polysaccharides using nuclear magnetic resonance (NMR) spectroscopy. Homogalacturonan (HG), consisting of 1,4-linked residues of α-D-galactopyranosyluronic acid (Gal*p*A), rhamnogalacturonan-I (RG-I), which contains a diglycosyl repeating unit with a strictly alternating sequence of 1,4-linked D-Gal*p*A and 1,2-linked L-rhamnopyranose (Rha*p*) residues in the backbone, and arabinan, were identified as the structural units of the obtained polysaccharides. HMBC spectra showed that arabinan consisted of alternating regions formed by 3,5-substituted and 1,5-linked arabinofuranose residues, but there was no alternation of these residues in the arabinan structure. Polysaccharide fractions scavenged the 1,1-diphenyl-2-picrylhydrazyl (DPPH) radical at 0.2–1.8 mg/mL. The correlation analysis showed that the DPPH scavenging activity of polysaccharide fractions was associated with the content of phenolic compounds (PCs).

## 1. Introduction

Plants of the Amaranthaceae family are associated with noxious garden weeds and ruderal plants. Perennial or annual herbaceous flowering plants of various species of the *Chenopodium* genus, known as the goosefoots, grow almost everywhere in the world and are among the most common cosmopolitan weeds. However, this family also contains valuable, useful plants. The genus of halophyte plant *Salsola* L. is one of the largest in the family Amaranthaceae. Plants of this genus are characterized by rapid regeneration, the ability to grow large biomass, resistance to high environmental temperature, tolerance to soil salinity and to extended drought conditions. Therefore, the role of plants of this genus is great in saline, arid regions of various countries with developed distant pastures. Over 150 species of the genus *Salsola* L., including annual semi-dwarf and dwarf shrubs and woody trees, are distributed in arid and semi-arid regions of the Middle East, Asia, Europe and Africa [1].

Extracts and decoctions of plants of this genus are used in world folk medicine to treat bacterial and viral, cardiovascular, skin diseases, coughs and flu, and in cosmetics [2]. Previously, several biologically active compounds were isolated from different types of Salsola: flavonoids, phenolic acids, saponins, triterpenes, lignans, sterols, fatty acids, alcohols, alkaloids, coumarins, as well as nitrogenous cyanogenic, isoprenoid and sulphur-containing compounds [3,4,5,6,7]. Most of these studies focus on phenolic compounds (PCs), which attract a lot of attention because of the great antioxidant activity.

Pectic polysaccharides are a family of complex polysaccharides present in all plant primary cell walls [8]. The irregular block sugar chains and various macromolecular segments of the linear and ramified regions characterize the complicated structure of pectic polysaccharides. The final model of the primary structure of the pectin macromolecule and the model of its biosynthesis have not been developed to date. The studies spanning the last 100 years have made it possible to establish the structure but not interlinking of the main domains of pectins. Homogalacturonan (HG) forms linear regions and the backbone of the substituted galacturonans (rhamnogalacturonan II, xylogalacturonan and apiogalacturonan). Rhamnogalacturonan (RG) forms the backbone of the RG-I, in which the arabinans, the galactans and/or the arabinogalactans form the side chains [9].

Pectin is resistant in the human stomach and small intestine and has to be fermented in the large bowel by colonic bacteria. Therefore, pectin belongs to dietary fibers and possesses good prebiotic properties [10]. Pectin is highly valued as a functional food ingredient because of hypolipidemic, hypoglycemic, satiating, antibacterial and antitumor effects [11]. In particular, pectic polysaccharides from various sources show antioxidant activity [12,13,14,15,16]. Various authors suggest that galacturonic acid (GalA) scavenges free radicals [17,18]; therefore, the HG domain mediates antioxidant activity [18,19]. Moreover, the antioxidant activity of pectins may also be associated with RG-I [20]. Additionally, the side chains of pectin may be feruloylated in certain cases, which might explain its considerable antioxidant potential [21].

Based on the wide distribution and ability to produce a large biomass, it is of interest to isolate pectin polysaccharides from plants of the genus Salsola L., since pectins make up the bulk of the plant cell and are its biologically active compounds. In this study, we assume pectins can be biologically active substances of saltwort *Salsola passerine*, which remains unexplored both in terms of polysaccharide composition and low-weight molecular phytochemicals and polyphenols. *S. passerina*, like other plant species of the genus Salsola, grows effectively in salt soil areas and can gain high biomass in semi-arid desert conditions, providing food for camels, sheep, goats, cattle. *S. passerina* is one of the most common species in Mongolia.

The present study aims to determine the structure and antioxidant activity of polysaccharides isolated from the semi-shrubs of *Salsola gemmascens* ssp. *passerina* (Bunge) Botschantz (main name—*Salsola passerina* Bunge) and to identify the component responsible for antioxidant activity.

## 2. Results and Discussion

### 2.1. Isolation and Characterization of Polysaccharides

Five extractants—cold water, hot water, water acidified to pH 2.0, 0.7% aqueous solutions of ammonium oxalate and 0.5% aqueous solutions of sodium carbonate—were successively used to extract polysaccharides from *S. passerine* (Figure 1). We performed the extraction with each extractant out until there were no sugars in the corresponding extract. As a result, eleven polysaccharide fractions were obtained. Fractions extracted with the sodium carbonate solution had the highest yield and those extracted with cold water—the lowest. Polysaccharides isolated with cold water containing a significant amount of Man, GalA and Ara residues were the principal components of polysaccharides extracted by other extractants.

**HW1**, **AC**, **OK1**, **SO1** were fractionated using a DEAE-cellulose (OH^−^) column. As a result, three polysaccharide fractions were obtained from each fraction by elution with 0.01, 0.1 and 0.2 M NaCl. The polysaccharides of the obtained fractions had a similar monosaccharide composition. The GalA and Ara residues were the principal components of their carbohydrate chains. The exception was the **HW1-1** fraction whose polysaccharides, similar to the polysaccharides of fractions extracted by cold water, were characterized by a significant content of Man and Glc residues (Table 1). This indicates that the extraction with cold water was incomplete, and a small part of the polysaccharides was extracted with hot water in the next step.

All parent fractions included protein components. Fractions extracted with sodium carbonate included the largest amount of protein (up to 39%). The largest part of the proteins was not connected to polysaccharides because it was removed during separation on DEAE-cellulose. However, a small part of the co-eluted protein seemed to be connected to polysaccharides.

### 2.2. NMR Spectroscopic Study 

Information about the structure of the main polysaccharides from *S. passerines* was obtained by a combined analysis of the NMR spectra of **SO1-1**, **SO1-2** and **SO1-3**. The NMR spectra of the three samples were similar (Figure 2, Figure 3, Figure 4 and Figure 5). The ^13^C NMR spectra of the samples (Figure 2) were assigned using ^1^H, ^13^C heteronuclear single quantum coherence spectroscopy (HSQC) spectra. Analysis of the ^1^H, ^13^C HSQC spectra (Figure 3, Figure 4 and Figure 5, Table 2) revealed substitutions in the monosaccharide residues based on the comparison of their ^13^C chemical shifts with those of the parent pyranoses and furanoses [22] and considering the glycosylation effects in the ^13^C NMR spectra of the carbohydrates [23,24], as well as data from our previous NMR studies of pectins [25]. The correlated spectroscopy (COSY), total correlation spectroscopy (TOCSY), rotating frame Overhauser effect spectroscopy (ROESY) and heteronuclear multiple bond correlation (HMBC) spectra revealed residues of α-D-galactopiranoside uronic acid (***GA*** in Table 2), α-L-rhamnopyranose (***R***) and α-arabinofuranose (***A***) in all three samples. Conclusions regarding monosaccharide composition, ring size and anomeric configuration were drawn based on the comparison of visible coupling constants and chemical shifts of the sugar residues and corresponding pyranoses [26,27] and furanoses [28,29].

The occurrence of 1,2-linked (label ***R***) and 2,4-substituted (label ***R′***) α-L-rhamnose residues in polysaccharides was confirmed by cross peaks at 1.25/17.9 ppm and 1.31/18.1 ppm in the ^1^H, ^13^C HSQC spectra (Figure 3, Figure 4 and Figure 5) and HMBC spectra (Figure 6 and Appendix A). The ROESY spectrum of **SO1-3** (Figure 7) included an inter-residue correlation peak of the anomeric proton of Rha residues and H-4 of GalA residues at δ_H/H_ 5.26/4.44 ppm, confirming the RG-I regions in polysaccharides.

Three intense signals at δ_H_ 5.16, 5.12 and 5.09 ppm belonging to terminal nonreduction arabinose residues (label ***A^T^***), 3,5-substituted arabinose residues (label ***A^S^***) and 1,5-linked arabinose residues (label ***A^L^***), respectively, were found in the anomeric region of ^1^H NMR spectra of **SO1-1** and **SO1-2** (Appendix A). In the anomeric region, the ^1^H NMR spectrum of **SO1-3** signals of 3,5-Ara and 1,5-Ara overlaps the intense signal belonging to the 1,4-linked D-galacturonic acid residues (label ***GA***) at δ_H_ 5.08 ppm.

The resonance of C-6 at δ_C_ 176.0 ppm indicated the predominance of non-methyl-esterified α-1,4-linked D-GalA residues in the structure of polysaccharides **SO1-3** (Figure 2c), but the signal of low intensity at δ_H/C_ 3.86/54.4 ppm confirmed that some GalA residues were methyl esterified.

The following inter-residue correlations H-1(glycosylating **GA**)/H-4(glycosylated **GA**) at δ_H/H_ 5.08/4.44 ppm in the ROESY spectrum of **SO1-3** (Figure 6) and H-4(glycosylated ***GA***)/C-1(glycosylating ***GA***) at δ_H/C_ 4.44/100.3 ppm in the HMBC spectrum of **SO1-3** (Appendix A) indicated on the galacturonan (HG) in the studied polysaccharides.

The correlation peak at δ_H/C_ 2.09/21.56 ppm in the HSQC spectrum of **SO1-3** (Figure 5) confirmed the *O*-acetylated residues in the structure of polysaccharides from *S. passerina*. No clear evidence was obtained for the attachment of the *O*-acetyl group to specific residues, since the intensity of their signals was low. Rha and GalA residues may be acetyl esterified [30]. The signals of O-acetyl groups are present only in the spectrum of sample **SO1-3**, which included polysaccharides with a high content of GalA, which may indirectly indicate the O-acetylation of GalA residues.

The resonance of C-6 at δ_C_ 176.0 ppm indicated the predominance of non-methyl-esterified α-1,4-linked D-GalA residues in the structure of polysaccharides (Figure 2).

The sequence of Ara residues in the repeating units was determined using the H/C correlations in the HMBC spectra and the H/H correlations in the ROESY spectra.

The inter-residue correlation peaks—H-1(glycosylating ***A^S^***)/C-5(glycosylated ***A^S^***) and H-1(glycosylating ***A^L^***)/C-5(glycosylated ***A^L^***) in HMBC spectra (Figure 7 and Appendix A) and H-1(***A^T^***)/H-3(***A^S^***) in ROESY spectra (Figure 6, Appendix A)—showed the side chains formed by single arabinose residues.

The average length of branches in the arabinan side chains, derived from the relative amounts of terminal and branched arabinose residues, was equal to one, confirming that the branches in the arabinan side chains comprised a single arabinose residue.

The ratio of ***A^T^***, ***A^S^*** and ***A^L^*** was approximately 1:1:4 in the ^1^H NMR spectrum of **SO1-1** and indicated that the lengths of the linear regions were four times the lengths of the branched regions.

The ratio of Ara residues in the spectra of **SO1-2** and **SO1-3** was not determined because of the overlap of the signal of the anomeric proton ***A^L^*** with the signal of the anomeric proton of ***GA*** in the ^1^H NMR spectrum (Appendix A).

Clear inter-residue correlation peaks in the HMBC spectra (Figure 7 and Appendix A) at δ_H/C_ 5.09/68.3 and 5.12/67.9 ppm for H-1(glycosylating ***A^L^***)/C-5(glycosylated ***A^L^***) and H-1(glycosylating ***A^S^***)/C-5(glycosylated ***A^S^***) mainly indicated that 3,5-Ara substituted 3,5-Ara, and 1,5-Ara substituted 1,5-Ara.

A possible structure of the repeating unit of the arabinan chain of polysaccharides from *S. passerines* is proposed below (Figure 1), where the lengths of the structural regions are arbitrary.

In addition, the following low-intensity peaks were found in the anomeric region of the ^1^H, ^13^C HSQC spectra: at δ_C/H_ 98.78/5.26 ppm belonging to Rha residues in the RG-I regions, at δ_C/H_ 103.40/4.54, 103.80/4.49, 104.68/4.48 ppm belonging to Gal residues, respectively (Figure 3 and Figure 4).

The occurrence of 1,2-linked (label R) and 2,4-substituted (label R’) α-L-rhamnose residues in polysaccharides was confirmed by the cross peaks at 1.25/17.9 ppm and 1.31/18.1 ppm in the ^1^H, ^13^C HSQC spectra (Figure 3, Figure 4 and Figure 5) and the HMBC spectra (Figure 7 and Appendix A). The ROESY spectrum of SO1-3 (Figure 6) included an inter-residue correlation peak of the anomeric proton of Rha residues, and H-4 of the GalA residues at δH/H 5.26/4.44 ppm confirmed the RG-I regions in polysaccharides.

Thus, three structural domains were identified in the polysaccharides isolated from *S. passerine*: arabinan, HG and RG-I. Considering the intensity of signals in the NMR spectrum, SO1-1 is dominated by the arabinan units, while SO1-3 is dominated by the galacturonan units. In the present study, no links between them were established. Nonetheless, it is possible that they represent domains of a complex pectin macromolecule.

Arabinans have been found in the cell wall of several plants and are believed to form RG-I side chains [31]. However, most of the evidence is based on co-extraction and/or co-elution of RG-I and arabinans [32,33,34]. Only a few studies found that the L-Ara residues are covalently attached to rhamnose residues at the *O*-4 position of the RG-I backbone [35,36].

1,5-linked residues of α-L-arabinofuranose form both the backbone and the side chains of most of the arabinans studied [37]. Backbone residues are usually substituted at *O*-2 and/or *O*-3 and/or at both positions, with *O*-3 substitutions predominating [32]. However, several arabinans with a high percentage of substitution at the *O*-2 position have also been found [38,39]. Other structures of arabinans have also been described. For example, in arabinans, both the furanose and pyranose forms of arabinose were found [40]. Terminal β-arabinofuranose residues may glycosylate 1,5-linked α-arabinofuranose residues of the backbone at position *O*-5 [41]. Various degrees of branching have been found, including single, linear and branched oligomeric and polymeric chains, with different linkage types. The almost linear 1,5-arabinan associated with protein was isolated from red wine [42]. The arabinans in pectins often have single substituted side chains [32,43]. Arabinans from soybean [44], apple [45], the inner bark of Norway spruce [46] were found to have a highly branched structure. Arabinan-rich pectins, which constituted 50% of the total pectic polysaccharides, have been obtained from pea *Pisum sativum* L. [47].

The roles of arabinans in plant cell walls remain unclear. It was established that arabinans can be substituted by terminal phenolic esters, particularly feruloyl or coumaroyl esters. Ferulic acid groups may be ester linked to *O*-2 of the arabinose residues [48,49]. Feruloyl esters may determine guard cell wall flexibility by providing the cross-links between arabinans and other wall polymers; this testifies a unique role for arabinans in determining the physical and functional properties of guard cell walls [50].

### 2.3. DPPH Radical-Scavenging Activity

Polysaccharide fractions from *S. passerine* scavenged the DPPH radical at concentrations of 0.2–1.8 mg/mL. The half-maximal DPPH inhibitory concentration (IC_50_) of them is given in Table 3. **CW2**, **CW3**, **SO1** and **SO2** demonstrated the highest activity, which exceeded 2.51–2.96 times that of commercial apple pectin (AP) activity. **CW2**, **CW3**, **SO1** and **SO2** scavenged 67, 69, 55 and 67% of DPPH radicals at a concentration of 1 mg/mL. Other fractions were less effective and scavenged only 31–48% of DPPH radicals at a concentration of 1 mg/mL.

The DPPH radical scavenging assay is widely used to evaluate the antioxidant property of plant polysaccharides. The activity of **CW2**, **CW3**, **SO1** and **SO2** seems to be comparable to that of polysaccharides from cantaloupe rinds [51], hawthorn wine pomace [52], fruit bodies of *Tremella fuciformis* [53] and apple pomace [54]. It should be noted that some other polysaccharides demonstrated the same level of DPPH scavenging activity at lower concentrations. These include pectins from *Chaenomeles sinensis* fruits [55], *Epilobium angustifolium* L. [56], *Thymus quinquecostatus* Celak. leaves [57], *Gardenia jasminoides* J. Ellis flowers [58] and *Ziziphus jujuba* cv. Muzao [59].

The DPPH radical scavenging activity of the fractions obtained by DEAE-cellulose elution was compared with the activity of the parent fraction **SO1** (Figure 8). The polysaccharides **SO1-1**, **SO1-2** and **SO1-3** obtained were less effective (*p* < 0.05) than the parent fraction **SO1**, exhibiting IC_50_ equal to 3.64 ± 0.18, 5.70 ± 0.92 and 5.70 ± 1.30 mg/mL, respectively.

On the basis of the yield and content of PCs in **SO1-1**, **SO1-2**, **SO1-3** and **SO1**, most of the PCs providing antioxidant activity were removed by anion exchange chromatography, assuming that they were not bound to the polysaccharide chains. The sum contents of PCs in polysaccharides **SO1-1**, **SO1-2**, **SO1-3** included about 18% from the content of PCs in the parent fraction **SO1**. It was detected that three fractions obtained on DEAE-cellulose provided only 24% of the DPPH radical scavenging activity of **SO1**, although they represented about 70% of the parent pectin (Table 1). This suggests that the antioxidant activity of **SO1** was mainly provided by the associated PCs but not by polysaccharides.

The relationship between the chemical characteristics of polysaccharides and DPPH scavenging ability was further investigated using correlation analysis. The total content of sugars and the (Ara + Gal)/Xyl ratio correlated negatively, whereas the content of PCs, Gal and Man, as well as PDI correlated positively with DPPH scavenging activity (Table 4).

We tested the five regression models, subsequently removing the less significant factors (according to the *p*-value). The linear regression, including the contents of PCs and Man as independent variables, resulted in the best model for prediction (adj. R^2^ = 0.82, *p* = 0.000) (Table 5). The content of PCs was the only factor contributing significantly to DPPH scavenging activity (*p* = 0.000, *β* = 0.79).

Thus, the correlation analysis showed that the DPPH scavenging activity of the sample from *S. passerine* is associated with the content of PCs. This is consistent with the results of Ref [54], whose authors evaluated the activity of apple pectins, and our previous study on fireweed pectins [56]. It is known that PCs may bind covalently to the side chains of RG I through the Ara and Gal residues and may be involved in the cross-linking of macromolecules [60].

It was shown that the removal of PCs from polysaccharides reduces the antioxidant activity but does not completely abolish it [60]. Several authors suggest that the antioxidant activity of pectins may be due to the hydroxyl and carboxyl groups of GalA residues [52,61]. Previously, we showed that the antioxidant activity of fireweed pectins is partly related to the xylogalacturonan chains [56]. However, in the present study, we failed to find the polysaccharide chains responsible for the DPPH radical scavenging activity of Salsola pectins. The small sample size (n = 14), which determines the statistical power of multiple regression [62], may be the reason we failed to identify the polysaccharide chains that contribute to the antioxidant activity of Salsola pectins.

## 3. Materials and Methods

### 3.1. Materials

Biological material: plant material, consisting of yellow-green annual branches with spherical dwarf leaves, was collected in August 2019 from the semi-shrubs of *Salsola gemmascens* ssp. *passerina* (Bunge) Botschantz. = *Caroxylon passerinum* (Bunge) Akhani et E.H. Roalson (main name—*Salsola passerina* Bunge) growing in Mandal-ovoo soum, Ömnö-Govi province, Mongolia. They were identified by Prof. B.Oyuntsetseg (School of Arts and Sciences, National University of Mongolia). The plant material was washed with distilled water and dried with filter paper.

The chemicals used are described in the Supplementary (Appendix B).

### 3.2. Isolation of Polysaccharides of S. passerina

Polysaccharides from the plant material were sequentially extracted, as described below; the extraction scheme is shown in Figure 1. At each stage, an exhaustive extraction of polysaccharides was carried out until the absence of reaction of the extract to the carbohydrate; the extraction mixtures were mixed in a mechanical stirrer.

Freshly picked plant material (234 g) was milled in a blender, distilled water (1 L) was added, and the resulting mixture was stirred in a mechanical mixer at 20 °C for 3 h. The mixture was centrifuged, and the residue of the plant material was treated again; the treatment was repeated three times. In the next stage, polysaccharides from the residues of plant materials were extracted with hot water at 80 °C for 3 h. The extraction was repeated twice (each time, the volume of added water was 1 L). Finally, the five aqueous extracts (three obtained with cold water (**CW1**, **CW2**, **CW3**) and two with hot water (**HW1**, **HW2**)) were obtained. Next, polysaccharides were extracted with acidified water (pH 2.0, 1 L) at 50 °C for 3 h. As a result, one extract (**AC**) was obtained. Next, polysaccharides were extracted with aqueous solutions of ammonium oxalate (0.7% *w*/*v*) at 70 °C for 6 h. The extraction was repeated three times (the first volume of salt solution added was 2 L; the second and third volumes were 1 L). Finally, the three extracts (**OK1**, **OK2**, **OK3**) were obtained. Next, polysaccharides were extracted with aqueous solutions of Na_2_CO_3_ (0.5% *w*/*v*) containing NaBH_4_ at 70 °C for 3 h. The extraction was repeated twice (the first volume of soda solution added was 3 L, the second—2 L). The two extracts (**SO1**, **SO2**) were obtained.

The carbohydrate content of each extract was detected using a phenol-sulfuric acid assay [63].

All extracts were dialyzed against distilled water for 48 h at 10 °C. Extracts **SO1**, **SO2** were previously acidified with a diluted solution of acetic acid to pH 5.6. The dialyzed extracts were concentrated on a Heidolph 4002 rotary evaporator (Germany) under reduced pressure at 40 °C.

Polysaccharides were precipitated from the extracts with a four-fold volume of 95% ethanol, centrifuged, washed twice with 95% ethanol, dissolved in distilled water, frozen and lyophilized. The yields of the polysaccharide fractions obtained are expressed in % (*w*/*w*) of mass of dry plant material and are presented in Table 1.

### 3.3. Ion Exchange Chromatography of Polysaccharide Fractions

The major polysaccharide fractions **HW1**, **AC**, **OK1**, **SO1** were separated on a DEAE-cellulose (OH-) column (2.5 cm × 40 cm). Each polysaccharide fraction (100 mg) was dissolved in 5 mL of 0.01 M NaCl, and the solution was applied to the column. The column was stepwise eluted with 0.01, 0.1, 0.2, 0.3, 0.5 and 1.0 M NaCl solution (400 mL of each eluent) at a flow rate of 0.9 mL/min. The fractions were collected at 12 min intervals using a low-pressure system Pharmacia Biotech (Sweden) with a FRAC-100 fraction collector, P-50 pump. The carbohydrate content in each tube was determined by the phenol–sulfuric acid method [63]. When separating each of the **HW1**, **OK1**, **SO1**, three major polysaccharide fractions were obtained (eluted with 0.01, 0.1 and 0.2 M NaCl). When separating **AC**, the fraction eluted 0.2 M NaCl was obtained as minor. In addition, minor fractions were obtained from all fractions by elution with 0.3, 0.5 and 1.0 M NaCl.

The separation procedure was repeated twice for **HW1**, **OK1** and four times for **AC**, **SO1**. Data on the monosaccharide composition and the yield of the fractions are presented in Table 1 as a mean of these experiments.

### 3.4. General Analytical Methods

The content of uronic acids was determined as described earlier [64,65]. The quantitative determination of protein was calculated using the Bradford method [66]. The quantitative determination of phenolics was performed with the Folin–Ciocalteu reagent using gallic acid as a standard [67]. The content of neutral monosaccharides was determined by gas–liquid chromatography (GLC), as described earlier in detail [68]. The sugar concentration was determined at 490 nm using the phenol–sulfuric acid assay [63].

The relative molar mass distributions (RMM) (including Mn, Mw and PDI) of the polysaccharide samples were determined by size exclusion chromatography with high-performance liquid chromatography (HPSEC); the procedure was described in detail earlier [69].

### 3.5. Nuclear Magnetic Resonance Spectroscopy

All homo- and heteronuclear NMR experiments of the samples were carried out on a Bruker Avance 600 spectrometer (Germany) at a probe temperature of 303, 313 and 318 K, which provided a minimum overlap of the signal of deuterated water with the polymer signals. The procedures for preparing the polysaccharide samples and the conditions of the NMR experiments were described earlier [69].

### 3.6. Antioxidant Activity

The DPPH solution (0.2 mM, in ethanol) was added to the pectin solution (0.4–3.6 mg/mL water) in equal proportions (*v*/*v*) and mixed. After incubation at 25 °C for 1 h, the absorbance of the sample was measured at 517 nm. The scavenging activity of the pectins was measured at four different concentrations, and the half-maximal inhibitory concentration (IC_50_, mg/mL) values were calculated based on a polynomial regression curve [70].

### 3.7. Statistical Analysis

The significance of the difference among the means in determining the antioxidant activity was estimated with one-way analysis of variance (ANOVA) and Fisher’s least significant difference (LSD) post hoc test at *p* < 0.05. The relationship between the chemical characteristics and activity of polysaccharide fractions was evaluated by the calculation of the Pearson correlation coefficients and multiple linear regression analysis. All calculations were performed using the statistical package Statistica 10.0 (StatSoft, Inc., USA). The data were expressed as the means ± s.d. of three independent experiments.

## 4. Conclusions

Polysaccharide fractions isolated from *S. passerine* with water and aqueous solutions of ammonium oxalate and sodium carbonate were characterized by a similar composition, including polysaccharides, protein and PCs. HG, RG-I and arabinan with regions formed by 3,5-substituted and by 1,5-linked arabinose residues were identified as the principal units of the polysaccharides obtained. Polysaccharide fractions of *S. passerine* demonstrated a moderate antioxidant potential. Fractions isolated with cold water and sodium carbonate scavenged the DPPH radical in vitro to a much greater extent than commercial apple pectin. The correlation analysis of the composition and activity of polysaccharide fractions obtained by anionic exchange chromatography revealed that the antioxidant capacity of polysaccharides of *S. passerine* is mainly due to the associated PCs.

## Data Availability

The data that support the findings of this study are available from the corresponding author upon reasonable request.

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
