# Peer review of "Polysaccharides of Salsola passerina: Extraction, Structural Characterization and Antioxidant Activity"

_ijms, 2022, doi:10.3390/ijms232113175_

Round 1

Reviewer 1 Report

The manuscript is suitable for publication with some minor modifications as the following:

1.   Page 1, line 14: What authors mean by "The ground part of the S. passerine"

2.    The NMR data of structure of the main polysaccharides (SO1-1, SO1-2 and SO1-3) should be clearly discussed with drawing the backbone or basic skeleton of the chemical structure for each unit of the three polysaccharides.

3.  In page 8, lines 152-154: The low intensity peaks at “δC/H 103.40/4.54, 103.80/4.49, 104.68/4.48 ppm” are observed in Figures 3 and 4 not in Figure 5.

4.   In page 8, lines 165-167: The following paragraph is not clear “The presence of signals of O-acetyl groups in the spectrum of sample SO1-3 with high content of GalA, and their absence in the spectra of samples SO1-3 and SO1-2 with low content of GalA, may indirectly indicate O-acetylation of GalA residues.”

5.      In page 8, line 169: “in the structure of polysaccharides (Figure 2).” should be “in the structure of polysaccharides of SO1-3 (Figure 2c).”

In addition to the following:

Page 1, line 14: "S. passerine" should be "Salsola passerine"

Page 1, line 26: “3,5-substituted arabinose residues and 1,5-linked arabinose residues” should be “3,5-substituted arabinofuranose residues and 1,5-linked arabinofuranose residues”

Page 1, lines 35, 40: "Amaranthaceae" should be non-italic

Page 7, Fig. 7, line 138: "carbon atoms” should be “proton atoms”

Page 7, line 140: "α-galactopiranosid” should be "α-D-galactopyranoside”

Page 7, line 140: "α-ramnopyranose” should be "α-L-rhamnopyranose”

Page 8, line 148: "1H spectra”   should be "1H NMR spectra”

Page 8, lines 150, 180: "1H spectrum”   should be "1H NMR spectrum”

Page 12, line 302: "(main name Salsola passerina Bungeand)" should be "(main name Salsola passerina Bunge)"

Page 12, lines 333-336: What authors mean by "The dialyzed extracts concentrated on a Heidolph 4002 rotary evaporator (Germany) under reduced pressure at 40°C. After dialysis, the extracts concentrated on a Heidolph rotary evaporator (Germany) under reduced pressure at 40°C."

Reviewer 2 Report

In the introduction description of the plant is very less informative. Write some more about of this species (Salsola passerine) and its composition and medicinal uses.

All abbreviations used in the method should be written in full when used for the first time.

The introduction is too scanty.

The authors declare that the phenolic compounds are responsible for the antioxidant capacity, in the bibliography we find the same statement. Why didn't the authors detect and then quantify the total phenols in each extract? Give reason.

In table 1, there is a column titled as PCs, but what does it refer to? it has no units. If it is the concentration of phenols, then in the methods they must include how they quantified the PCs.

If the authors did such a thorough job identifying carbohydrates, why didn't they isolate, purify, and identify phenolic compounds?

Round 2

Reviewer 2 Report

The authors responded to all comments and added sufficient information to the manuscript.